# Refactoring and Optimization of Bridge Dynamic Displacement Based on Ensemble Empirical Mode Decomposition

**DOI:** 10.3390/s19143125

**Published:** 2019-07-15

**Authors:** Yingquan Zou, Yunpeng Chen, Peng Liu

**Affiliations:** 1School of Information Science and Technology, Southwest Jiaotong University of China, Chengdu 611756, China; 2State key Laboratory of Rail Transit Engineering Informatization, China Railway First Survey and Design Institute Group, Xi’an 710043, China

**Keywords:** bridge, vibration, dynamic displacement, accelerometer, EEMD

## Abstract

Considering the lack of precision in transforming measured micro-electro-mechanical system (MEMS) accelerometer output signals into elevation signals, this paper proposes a bridge dynamic displacement reconstruction method based on the combination of ensemble empirical mode decomposition (EEMD) and time domain integration, according to the vibration signal traits of a bridge. Through simulating bridge analog signals and verifying a vibration test bench, four bridge dynamic displacement monitoring methods were analyzed and compared. The proposed method can effectively eliminate the influence of low-frequency integral drift and high-frequency ambient noise on the integration process. Furthermore, this algorithm has better adaptability and robustness. The effectiveness of the method was verified by field experiments on highway elevated bridges.

## 1. Introduction

The dynamic displacement response of a bridge contains valuable structural behavior information, which can effectively reflect the structural behavior characteristics (natural frequency, modal shape, damping ratio, etc.) and is often applied to various structural health monitoring (SHM) and structural control (SC) methods [1,2]. Under the action of traffic load, the dynamic displacement response of a bridge is important for evaluating the impact of vehicles on the performance of the road structure. Dynamic displacement response under live load is an important factor in leading fatigue damage. Accurate evaluation of this dynamic displacement response is important for ensuring the effective maintenance of steel bridges [3,4].

Dynamic displacement measurement methods are mainly divided into direct and indirect displacement measurement method. Traditional direct displacement measurements include linear variable differential transformers (LVDTs), laser Doppler vibrometers (LDVs), and vision-based systems. Although the methods have high measurement accuracy, they require a reference point, and in practical applications, it is often difficult to find a suitable reference point at the test site [5,6,7]. Differential global positioning system (DGPS) technology solves this problem by installing a receiver as a reference station at a fixed location. With the plane displacement measurement accurate to the submillimeter, DGPS technology has become a mature measurement method in geodetic surveying and a standard method for monitoring the movement of bridge structures [8]. However, this technique has poor accuracy in vertical displacement measurements and cannot meet the accuracy requirements of bridge dynamic displacement measurements [9].

Regarding indirect displacement measurement methods, the accelerometer has been widely used due to its wide frequency band response, small size, easy installation, and low cost [10]. The dynamic displacement is calculated by quadratic integration of the acceleration signal of the bridge vibration collected by the accelerometer. The integral calculation of the displacement is mainly divided into two types: the frequency domain integral and the time domain integral. For frequency domain integration, the acceleration time domain signal is first transformed into the frequency domain by fast Fourier transform (FFT). Then, using the integral property in the frequency domain, the acceleration frequency domain signal is subjected to a second integral calculation. Finally, the displacement time domain signal is obtained by inverse-frequency inverse fast Fourier transform (IFFT) of the frequency domain signal obtained by integration. In order to eliminate the influence of low-frequency trend terms and high-frequency noise on the integration process, filtering is usually performed in the frequency domain. However, frequency domain filtering has the problem of cutoff frequency selection and truncation error, and it is easy to generate a large displacement calculation error when the original signal spectrum is unknown [11]. Time domain integration avoids problems such as cutoff frequency selection, but displacement signals obtained by quadratic integration in the time domain using a micro-electro-mechanical system (MEMS) accelerometer can be severely offset or even lead to erroneous results. Because the output signal of the MEMS accelerometer has interference factors such as the DC component, zero drift, long-period interference, and temperature drift, the error caused by these factors will be amplified with the integral [12,13]. Therefore, studying how to remove these trend terms in the acceleration signal is an important part of time domain integration.

Kandula et al. [14] proposed an acceleration signal model that is the sum of exponentially damped sinusoidal signals. The displacement response is obtained by modeling the noiseless acceleration and then integrating it twice. This method can ignore the boundary conditions of the initial acceleration and initial velocity and has high calculation accuracy. However, this method involves the matrix inverse operation of a large data set, which has high complexity and long calculation time. Lin et al. [15] used the vibration law of the structure to find the zero point of velocity and displacement in the steady-state vibration phase. This method solved the integral boundary value problem and finally used trend term processing to eliminate the influence of integral drift. Thai et al. [16] used FFT frequency domain filtering to achieve accelerometer measurement displacement. FFT Filter is known to provide a higher accuracy compared to infinite impulse response (IIR) Filter and finite impulse response (FIR) Filter. However, the integration process is performed in the frequency domain, and there are problems of frequency truncation error and cutoff frequency selection. Chen et al. [17] uses empirical mode decomposition (EMD) method to extract integrated signals trend, discussed the advantages of EMD method in this case, proves that EMD has a good application in integrated signal trend extraction. However, due to the characteristics of EMD decomposition itself, mixing and an end effect are easy to occur. Shi et al. [18] proposed an integral transformation method based on FFT time-frequency conversion and EMD adaptive filtering, which provides a new way to obtain displacement signals. Although compared to the literature [16,17], the effect has improved. However, this method still has risks such as truncation error, endpoint effect and modal mixing. Jiang et al. [19] proposed a hydraulic pump fault identification method, which uses the ensemble empirical mode decomposition (EEMD) and EMD methods to decompose and reconstruct the vibration data of the pump casing. Through short-time maximum entropy spectrum analysis, the intrinsic mode function (IMF) component which is most sensitive to the fault is selected, and the fuzzy C-means clustering algorithm is used to identify the fault pattern. The recognition results show that compared with EMD, the number of iterations of the EEMD method is greatly reduced, and the fault recognition rate is also significantly improved.

In this study, the EEMD method was used to successively decompose and reconstruct the vibration acceleration signal, velocity signal, and displacement signal of a bridge, which effectively eliminated the interference caused by the high-frequency noise and low-frequency trend in the integral calculation process, and obtained the dynamic displacement signal that satisfied the calculation accuracy.

## 2. Acceleration Integral and EEMD Algorithm

### 2.1. Acceleration Integral Calculation Displacement Principle

Let the displacement be the second integral relationship of acceleration:
(1)$$s=s_{0}+v_{0}t+\int_{0}^{t}\int_{0}^{t}adt$$
where $s_{0}$ represents initial displacement; $v_{0}$ represents initialising speed.

In order to facilitate processing calculations in the computer, the integral discrete form uses a trapezoidal integral formula:
(2)$$\int_{t\left(0\right)}^{t\left(n\right)}x\left(t\right)dt\approx \sum_{i=1}^{n}\left(\frac{x\left(i-1\right)+x\left(i\right)}{2}\right)\Delta t$$
where x(t) represents integral item; t(0) represents integration start time; t(n) represents point termination time; Δt represents sampling interval.

Equation (2) is substituted into Equation (1) to obtain the displacement integral calculation formula:
(3)$$s\left(n\right)=s\left(0\right)+nv\left(0\right)\Delta t+\left[\frac{2n-1}{4}a\left(0\right)+\left(n-1\right)a\left(1\right)+\cdots +a\left(n-1\right)+\frac{1}{4}a\left(n\right)\right]\Delta t$$
where s(0) represents initial displacement; v(0) represents initialising speed; a(0) represents initial acceleration; a(1)–a(n) represents sampled n acceleration sample points; s(n) represents displacement signal obtained by integrating; Δt represents acceleration signal sampling interval.

Δt is an important parameter for calculating the displacement, which repr-esents the acceleration sampling time interval, and the sampling rate is reciprocal. When Δt is smaller, the sampling rate is higher, and the result of the integral operation of Equation (3) is more similar to the numerical operation result. However, in practical applications, many factors such as calculation cost and acquisition equipment determine an optimal sampling rate interval. In this test, the sampling rate was 1 kHz, and the accuracy requirement was guaranteed.

As shown in Equation (3), the displacement is related to the initial displacement and the initial velocity. The initial displacement affects the final displacement result in the form of a constant superposition, and the initial velocity is the result of the final displacement with time as a linear function. As time increases, the cumulative error increases. In addition, the low frequency interference caused by the MEMS accelerometer itself and the zero drift of the acquisition system is amplified by integration. Therefore, filtering processing such as low-pass filtering and smoothing filtering is usually used to reduce the influence of the abovementioned phenomenon. However, the limitation of the filtering process is too large, and the cutoff frequency is difficult to select. Therefore, an adaptive processing method is needed to reduce these interferences, and EEMD is an empirical data analysis method. It is decomposed according to the time characteristic scale of the signal itself, and each IMF component has different time feature scales. Therefore, it can be screened according to different time feature scales, thereby filtering out unnecessary components and achieving the purpose of adaptive filtering.

### 2.2. EEMD Algorithm

EEMD is developed on the EMD algorithm, which is the core algorithm of the Hilbert–Huang transform (HHT) [20]. The purpose of the EMD algorithm is to decompose non-stationary and nonlinear signals into a set of steady-state and linear IMFs, and the IMF satisfies the following two properties:
The extreme point $N_{e}$ and the zero crossing $N_{Z}$ of the signal are equal or at most one different:
(4)$$\left(N_{Z}-1)\leq N_{e}\leq (N_{Z}+1\right)$$
At any point, the mean of the envelope $S_{\max}\left(t\right)$ defined by the local maxima and the envelope $S_{\min}\left(t\right)$ defined by the local minima is zero:
(5)$$\left[S_{\max}\left(t\right)+S_{\min}\left(t\right)\right]/2=0$$



However, there are problems and deficiencies in EMD. The IMF components obtained by EMD decomposition have modal mixing and an end effect. Modal mixing is reflected in different IMF components with the same scale information. At this time, the low frequency component may be mixed with the effective information, which makes the selection of the IMF component difficult, and the signal is seriously shifted after the integration. The endpoint effect is accompanied by each screening process, which gradually enlarges as the operation runs, thus affecting the entire data sequence. In order to overcome these problems, Huang et al. [21] proposed the EEMD method. The EEMD decomposition principle is as follows: Add white noise sequences with different frequencies to the target data. When the additional white noise is evenly distributed in the entire time–frequency space, the time–frequency space is composed of different scale components divided by the filter bank. Signals of different scales are automatically projected onto the appropriate reference scale established by white noise, which can effectively solve the modal mixing phenomenon and the end effect. Although each test may produce very noisy results, in the case where the test sample is large enough, the noise will almost disappear after the average calculation [22]. As an improvement of the EMD algorithm, EEMD includes the following steps:
Add a white noise sequence to the target data:
(6)$$X\left(t\right)=x\left(t\right)+n_{i}\left(t\right)$$
Decompose the white noise-added data into multiple IMF components:
(7)$$X\left(t\right)=\sum_{j=1}^{K}c_{j}\left(t\right)+r_{K}\left(t\right)$$
Repeat steps (1) to (2), and add a white noise sequence different from (1) to the target data:
(8)$$X_{i}\left(t\right)=\sum_{j=1}^{K}c_{ij}\left(t\right)+r_{iK}\left(t\right)$$
Obtain the (total) mean of the corresponding multiple IMFs as the final result:
(9)$$c_{j}=\frac{1}{N}\sum_{i=1}^{N}c_{ij}\left(t\right)$$
where $c_{j}$ represents the IMF component obtained by EEMD decomposing the signal after adding white noise sequence; $r_{K}\left(t\right)$ represents the residual of the signal EEMD after adding white noise sequence; $c_{ij}\left(t\right)$ represents the IMF component obtained by EEMD decomposing the signal after adding the *i*-th white noise sequence; $r_{iK}\left(t\right)$ represents the residual of the signal EEMD after adding the *i*-th white noise sequence; N represents the total number; K represents the number of IMF components.


The original signal is decomposed into multiple IMF components by EEMD. Each IMF represents an internal modal of the original signal. Therefore, we can select the modal components we need for analysis and processing, so as to achieve the purpose of adaptive data processing.

### 2.3. Displacement Reconstruction Algorithm

In the integral process of acceleration to displacement, the effective IMF component is continuously selected by the EEMD algorithm for reconstruction, and finally the dynamic displacement is obtained. Table 1 shows the displacement reconstruction algorithm.

Where a(t) represents input acceleration signal; $n_{i}\left(t\right)$ represents the *i*-th white noise sequence with different frequencies; $A_{i}\left(t\right)$ represents an acceleration signal of the *i*-th white noise sequence is added; $c_{ij}\left(t\right)$ represents the IMF component obtained by EEMD decomposing the signal after adding the *i*-th white noise sequence; $r_{iK}\left(t\right)$ represents the residual of the signal EEMD after adding the ith white noise sequence; $c_{j}$ represents the overall mean value of multiple IMFs after N times EEMD decomposition; $a_{1}\left(t\right)$ represents the acceleration obtained by the reconstruction; v(t) represents velocity signal obtained by integrating the reconstructed acceleration signal; $V_{i}\left(t\right)$ represents a speed signal of the *i*-th white noise sequence is added; $v_{1}\left(t\right)$ represents the velocity obtained by the reconstruction; s(t) represents displacement signal obtained by integrating the reconstructed velocity signal; $S_{i}\left(t\right)$ represents a displacement signal of the *i*-th white noise sequence is added; $s_{1}\left(t\right)$ represents the displacement obtained by the reconstruction; x(t) represents output displacement signal.

## 3. Experiment and Analysis

This section covers the experiments and analysis conducted on bridge analog signals, vibration table measured signals, and bridge measured signals.

### 3.1. Bridge Analog Signal Simulation Experiment

#### 3.1.1. Constructing Bridge Analog Signals

Generally, the frequency of the bridge dynamic response vibration signal is 1–10 Hz, so the ideal displacement signal constructed for this paper was:
(10)s=10sin(5πt)+10sin(10πt)


The acceleration a is the second derivative relation of the displacement s, so the ideal acceleration signal is
(11)$$a=-250\pi^{2}\sin(5\pi t)-1000\pi^{2}\sin(10\pi t)$$


The ideal acceleration signal was constructed in MATLAB, and a random white noise of Gaussian distribution was added on the basis of this, as shown in Figure 1. Noise-induced acceleration signal signal-to-noise ratio (SNR) is 17 dB.

In order to obtain the ideal displacement signal, the acceleration signal processing with noise was processed by four methods: frequency domain bandpass filtering [16], EMD adaptive filtering [17], FFT+EMD filtering processing [18] and EEMD adaptive filtering.

#### 3.1.2. Frequency Domain Bandpass Filtering

Figure 2 is a flow chart of the method based on frequency domain low pass filtering. First, the acceleration time domain signal was subjected to FFT processing and converted into a corresponding frequency domain signal. Then, an ideal bandpass filter was set in the frequency domain to filter the acceleration frequency domain signal.
(12)$$H\left(f\right)=\{\begin{array}{cc} 1 & f1\leq f\leq f2\\ 0 & else \end{array}$$
where H(f) represents ideal frequency domain bandpass filter.

Since the bridge vibration analog signal frequency was 2.5 Hz, f1 = 1 Hz and f2 = 10 Hz. The filtered frequency domain signal was subjected to quadratic integration in the frequency domain to obtain a shifted frequency domain signal. Then, the displacement frequency domain signal was subjected to IFFT processing to obtain a shifted time domain signal. As can be seen from Figure 3, due to the frequency truncation error and the like, there was a certain error between the signal at 0–1 s and 9–10 s and the ideal signal.

#### 3.1.3. Based on EMD Adaptive Filter Processing

Figure 4 is a flow chart of a method based on EMD adaptive filter processing. In the figure, a represents the sampled original acceleration signal, a′ represents the acceleration signal after the decomposition reconstruction process, v represents the integrated velocity signal, v′ represents the velocity signal after the decomposition reconstruction process, s represents the displacement signal obtained by the integration, and s′ represents the displacement signal after the decomposition reconstruction process.

As shown in Figure 5, the displacement result obtained by the EMD adaptive filter processing method almost coincided with the ideal signal at 1.5–9 s. However, due to the modal mixing phenomenon and the end effect, the other time periods were different from the ideal signal, and the overall effect was not satisfactory. Especially at the end of the signal, the distortion amplitude is more than six times.

#### 3.1.4. Based on FFT+EMD Filtering

As can be seen from Figure 6, the principle based on the FFT+EMD filtering processing method was similar to the method based on frequency domain filtering processing. Only an EMD adaptive filtering process was performed on the shifted time domain signal obtained by IFFT.

As shown in Figure 7, compared with the method based on frequency domain filtering and EMD-based adaptive filtering, the ideal displacement signal and the displacement signal obtained by the FFT+EMD filtering processing method had a longer coincidence time.

#### 3.1.5. Based on EEMD Adaptive Filter Processing

The principle based on the EEMD adaptive filtering method was similar to the EMD–based adaptive filtering method, except that EMMD was used instead of EMD. The flow chart based on EEMD adaptive filtering process is shown in Figure 8:

The number of ensembles is set 200, and the level of noise added is set 0.2. These parameters are uniform in the article. As shown in Figure 9, the displacement signal after the EEMD filtering process and the ideal displacement signal were highly coincident throughout the time period, and the overall effect was better than the previous three methods, which demonstrates the effectiveness of the EEMD adaptive filtering method.

#### 3.1.6. Error Analysis of Integral Results

The root-mean-square error (RMSe), kurtosis error (Ke), and phase error (Pe) of the abovementioned four methods with respect to the ideal signal were calculated separately, as shown in Table 2. 

The RMSe and Ke of the EEMD adaptive filtering method were both below 1.5%, and the Pe was less than 2°, which could meet the calculation accuracy. Compared with the other three methods, the EEMD adaptive filtering method had higher precision for the integral processing of the low-frequency acceleration signal.

### 3.2. Vibration Table Data Acquisition Test

A multi-channel vibration data synchronous acquisition system was used to collect the vibration acceleration signal of the vibration table, and the abovementioned four methods were used to reconstruct the dynamic displacement of the vibration table. The obtained displacement signal was compared with the displacement obtained by laser displacement sensor to verify the effectiveness of the displacement reconstruction method based on the EEMD adaptive filtering.

#### 3.2.1. Data Acquisition and Algorithm Verification Test Based on Vibration Table

Figure 10 is a physical diagram of the vibrating table. It can be seen from the figure that the test mainly uses two sets of data acquisition systems: acceleration acquisition system and displacement acquisition system. The acceleration acquisition system consists of an acceleration sensor, a multi-channel acceleration data acquisition system, a 12-DC power supply battery, and a host computer that receives data. The displacement acquisition system consists of a laser displacement sensor, a power supply, a display panel, a data logger, and a host computer that receives data.

The vibration acceleration signal was given by the vibration table. The vertical and horizontal two-dimensional variable frequency vibration control was realized by the vibration table frequency conversion controller controlling a pair of eccentric wheels to generate vibration simulation quantities with different dimensions and amplitudes. In the test, the vibration frequency was set to 8.7 Hz and the amplitude was 0.13 mm. The three-axis MEMS accelerometer sampled the acceleration of the vibrating table and transmitted the acceleration signal to the upper computer through the multi-channel acceleration acquisition system. The vibration table displacement reconstruction was realized by the abovementioned four methods on the upper computer platform, and different methods were compared and analyzed.

Figure 11 shows the shaker acceleration signal and the frequency domain spectrum. The figure shows the waveform data of the vibration acquisition system continuously sampling for 10s at a sampling rate of 1 kHz. The frequency domain spectrum showed that the frequency of the vibrating table signal was mainly concentrated at 8.7 Hz, which was consistent with the vibration frequency parameter set by the vibrating table. It can be visually seen from the spectrum of the acceleration signal that there was a large-amplitude, low-frequency component.

#### 3.2.2. Displacement Reconstruction of Vibration Table

From the time domain and frequency domain spectra in Figure 12, in addition to the EMD adaptive filtering method, the other three methods effectively filtered out the low-frequency components. The displacement signal obtained by the frequency domain filtering method was generally better, but due to the truncation error, there were some offsets at the beginning and end of the signal, exceeding the actual amplitude of the vibrating table.

Figure 13 is a partial IMF component obtained by decomposition of the acceleration signal EMD and its spectrum. It can be seen from the figure that the main frequency band of IMF4 and IMF5 was 8.7 Hz, which was consistent with the set vibration frequency. However, IMF4 and IMF5 had a modal mixing phenomenon and an end point effect, which caused the error integral to amplify and caused serious offset, which directly affected the effect of empirical modal analysis.

Figure 14 shows a partial IMF component obtained by decomposing the displacement signal EMD obtained by the frequency domain filtering process and its spectrum. It can be seen from the figure that IMF1 was the target signal. However, in the process of EMD decomposition, the end effect occurred and the degree of offset at the end of the displacement signal increased.

#### 3.2.3. Error Analysis of Integral Results

The root-mean-square error (RMSe), kurtosis error (Ke), and phase error (Pe) of the abovementioned four methods with respect to the ideal signal were calculated separately, as shown in Table 3. 

The RMSe and Ke of the EEMD adaptive filtering method were both below 0.5%, and the Pe was less than 2.2°, which could meet the calculation accuracy. Compared with the other three methods, the EEMD adaptive filtering method had higher precision for the integral processing of the low-frequency acceleration signal.

In summary, the displacement signal obtained based on the EEMD adaptive filter processing method was more accurate. From the acceleration to the speed, and then to the displacement, EEMD decomposition was used in order to eliminate the invalid components, which basically eliminated the effects of high-frequency noise, low-frequency trend terms, and DC components in the original shaker acceleration signal. The calculated dynamic displacement peak-to-peak value of the vibrating table was 0.26 mm, which was consistent with that given in the shaker manual. The validity of the reconstruction displacement of this algorithm was thus verified.

### 3.3. Bridge Field Data Acquisition Test

The effectiveness of dynamic displacement reconstruction based on the EEMD adaptive filtering method was verified by software simulation and vibration test bench. The dynamic displacement response of bridges is often used for bridge health monitoring and early warning. Therefore, it is of great practical significance and value to provide a basis for bridge health monitoring by analyzing and processing the vibration data of bridges.

#### 3.3.1. Data Acquisition and Algorithm Verification Test Based on Highway Elevated Bridge

This experimental test site uses a highway elevated arch bridge with a height of about 15 m and a length of about 800 m. By analyzing the force of the bridge structure, a three-axis MEMS acceleration sensor is installed at the top of the arc of the bridge. For arched bridges, the vibration at the top of the arc is most pronounced when the vehicle passes. 

Figure 15 is a schematic diagram of bridge site data collection environment. This figure shows in detail how the acceleration signal of the bridge vibration is obtained during the experiment. Figure 16 is a physical map of field data acquisition. The accelerometer uses an AKE390B-MEMS voltage output accelerometer. The accelerometer output signal is three-axis with a measurement range of ± 2 g. In this test, two acceleration sensors were installed at one measuring point. By comparing the two acceleration signals sampled, the validity of the data is guaranteed. The vibration data acquisition system is a 12-channel, high-precision data acquisition system independently developed by the project team. The maximum sampling speed is 250 kSPS and the sampling accuracy can reach 0.15 mV. The camera uses two 1080P wide-angle cameras to clearly record the passing vehicles on the bridge. There are three groups of 12 V batteries, which supply power for the acquisition system, industrial computer and portable TFT screen. The industrial computer is used for the temporary storage of data and the TFT screen is convenient for on-site debugging. The 4G router is connected to the industrial computer for remote control.

In combination with the video recorded by the camera, the dynamic displacement response of the bridge caused by heavy trucks and large slag trucks was selected. At this time, the dynamic displacement of the bridge is large, and the structural behavior characteristics obtained from it are more obvious. According to the dynamic displacement of the bridge, vehicle type, driving speed, rough road conditions, etc., the fatigue stress of short-span and mid-span bridges under dynamic traffic loads can be modeled to evaluate the fatigue reliability of the bridge [23].

Figure 17 shows the bridge vibration acceleration signal and frequency domain spectrum of a period of 60 s. It can be seen from the figure that the vibration acceleration signal main frequency of the bridge was 2.487 Hz, and there was a low-frequency DC component with a high amplitude. Therefore, the passband frequency was set to 1–10 Hz during the frequency domain filtering process.

#### 3.3.2. Displacement Reconstruction of Viaduct

The dynamic displacement reconstruction results of the bridge are shown below.

Figure 18 shows the dynamic displacement of the bridge with two peaks. Combined with the video taken by the camera, the first peak was the dynamic displacement response caused by a car, with a small peak and which quickly decayed to zero. The second peak was caused by a large fullload truck, with a large peak, up to 1.4 mm, and a free decay time, which lasted about 35 s. Using these data, bridge warnings can be made and some structural features of the bridge, such as the natural frequency and damping ratio of the bridge, can be obtained. It can be seen from Figure 18b that the dynamic displacement signal reconstructed based on the EMD adaptive filtering method was completely distorted due to the modal mixing phenomenon and the end effect.

Figure 19 is a comparison diagram of a displacement signal based on FFT+EMD filtering and a displacement signal based on frequency domain filtering. The displacement results obtained by the two as a whole were almost coincident. However, within 0–1 s, due to the end effect caused by EMD decomposition, the displacement signal based on FFT+EMD filtering had a small amplitude signal distortion.

As mentioned above, usually due to the influence of the frequency cutoff error, the results obtained by the FFT filtering method will have certain errors at the beginning or end. Since there is no vehicle passing the test point in the first five seconds and the last five seconds of the acceleration signal selected in Figure 18, the vibration displacement is close to zero. The author judges that the method based on FFT filtering may have a certain error in the first and last paragraphs in this case, but since the actual displacement signal is close to zero, even if there is a certain error, it is not obvious. In order to verify the author’s idea, the author intercepted the original acceleration signal of 21.5–40 s and performed FFT-based filtering on the acceleration signal. Since the acceleration signal is intercepted from the vicinity of the peak of the displacement, if the result obtained by the method based on the FFT filtering process has a certain error at both ends, it becomes very obvious. Since the displacement signals in Figure 18a,d are highly coincident over the entire time period, the displacement signal of Figure 18a at 21.5–40 s is taken as the ideal displacement signal. The result of the intercepted signal after the FFT filtering process is compared with “the ideal signal” as shown in Figure 20.

As can be seen from the Figure 20, due to the existence of factors such as the frequency cutoff error, there is a large amplitude distortion at the head end of the signal. Compared to the “ideal signal”, the degree of distortion reached 18.14%. Moreover, it can be seen from the frequency domain diagram that the method introduces a small amount of low frequency error.

In order to verify the effectiveness of the proposed EEMD-based adaptive filtering method, the “ideal displacement signal” is compared with the intercepted signal through the EEMD-based adaptive filtering process, as shown in the Figure 21.

It can be seen from the figure that the result based on the EEMD adaptive filtering process and the “ideal displacement” are highly coincident, which embodies the effectiveness of the proposed algorithm.

## 4. Conclusions

In this paper, the EEMD algorithm was used to reconstruct the vibration acceleration signal to the displacement signal. The vibration acceleration signal, the speed signal, and the displacement signal were corrected in turn and, finally, the displacement signal satisfying the calculation precision was reconstructed. The effectiveness of reconstructing the dynamic displacement of the bridge based on the EEMD adaptive filtering algorithm was verified by software simulation, vibration test bench verification, and a field test. Compared with the frequency domain filtering processing method, the EMD adaptive filtering processing method, and the FFT+EMD filtering processing method, the superiority of the proposed method was demonstrated. Combined with video recordings, the violent dynamic displacement response of bridges was found to be caused mostly by heavy or large trucks. Limiting the number of vehicles passing over a bridge and the speed of the vehicles can reduce the fatigue damage of a bridge structure and provide good protection for the bridge. However, the method proposed in this paper also has certain deficiencies. Through the simple beam hammer test, the authors found that the proposed method is not suitable for broadband excitation signals. The team next plans to make some improvements based on the method proposed in this paper, hoping to realize the reconstruction of the broadband excitation acceleration signal.

## Figures and Tables

**Figure 1 sensors-19-03125-f001:**
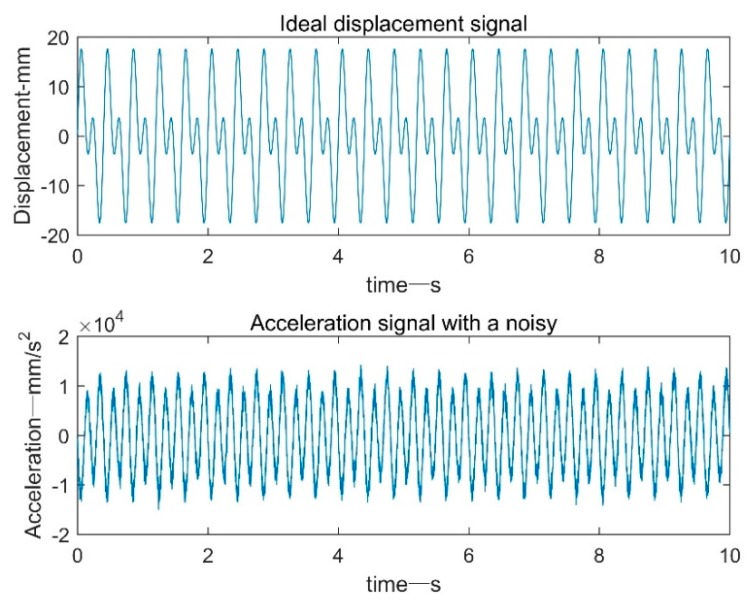
Acceleration signal diagram.

**Figure 2 sensors-19-03125-f002:**
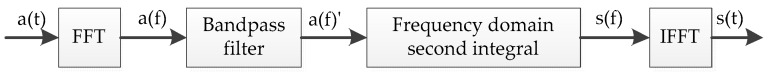
Flow field based on frequency domain bandpass process.

**Figure 3 sensors-19-03125-f003:**
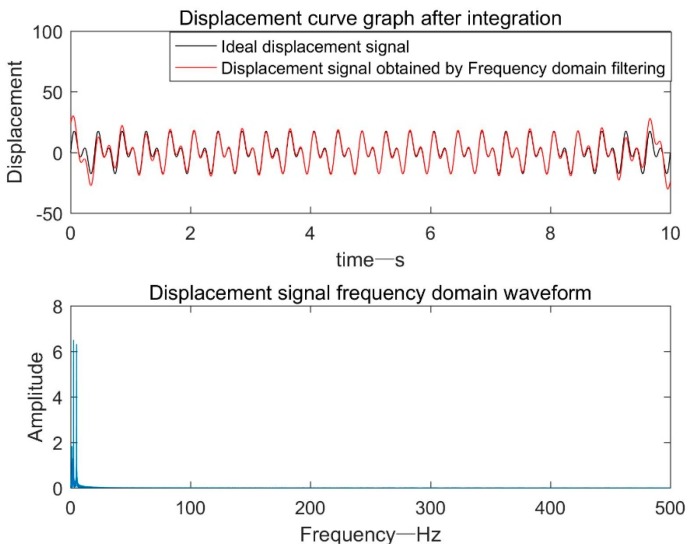
Displacement signal after frequency domain filtering.

**Figure 4 sensors-19-03125-f004:**
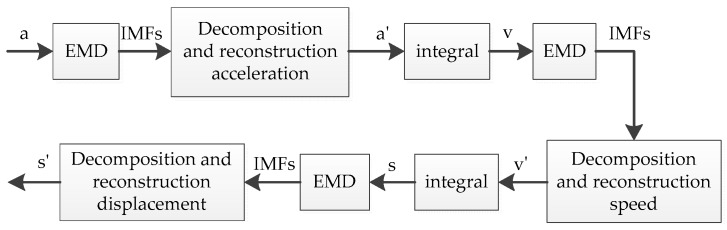
Flow field based on EMD adaptive filtering process.

**Figure 5 sensors-19-03125-f005:**
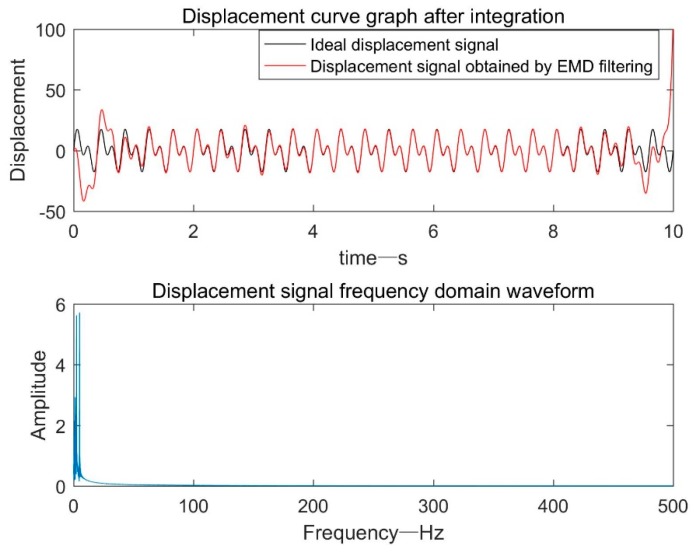
Displacement signal after EMD adaptive filtering.

**Figure 6 sensors-19-03125-f006:**
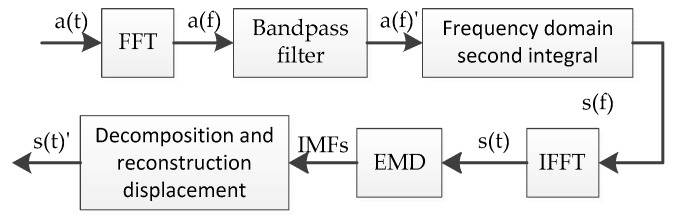
Flow field based on FFT+EMD adaptive filtering process.

**Figure 7 sensors-19-03125-f007:**
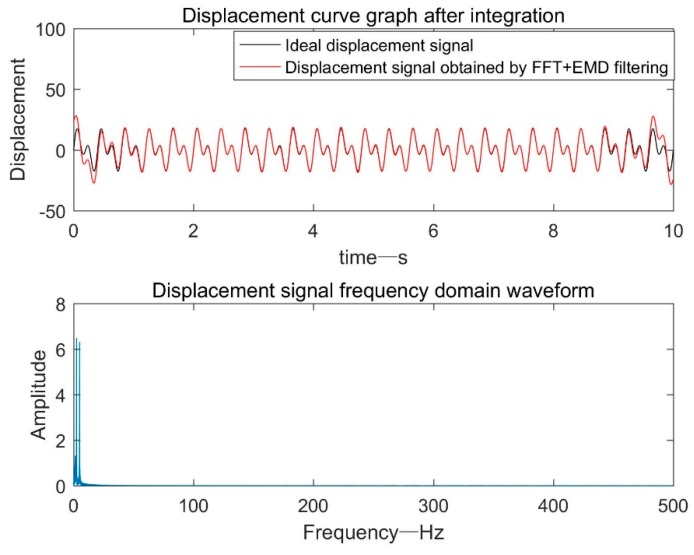
Displacement signal after FFT+EMD adaptive filtering.

**Figure 8 sensors-19-03125-f008:**
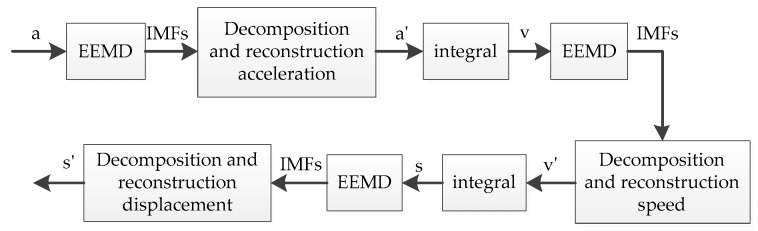
Flow field based on EEMD adaptive filtering process.

**Figure 9 sensors-19-03125-f009:**
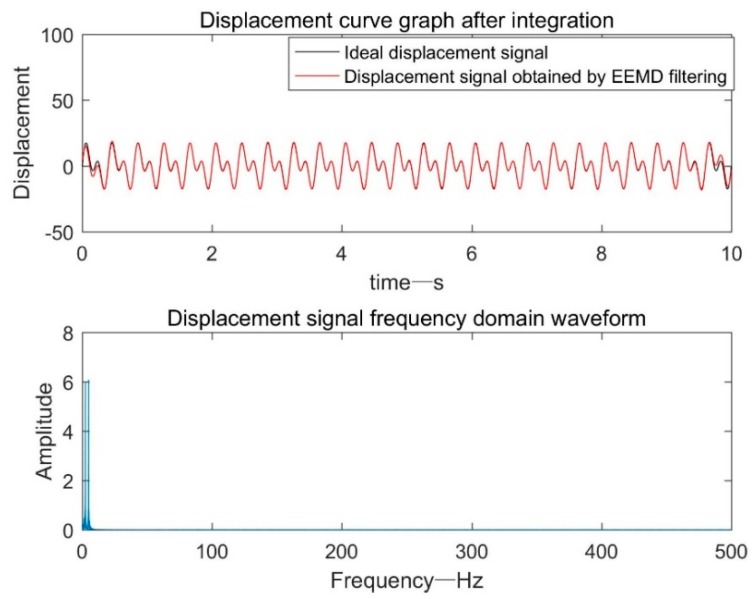
Displacement signal after EEMD adaptive filtering.

**Figure 10 sensors-19-03125-f010:**
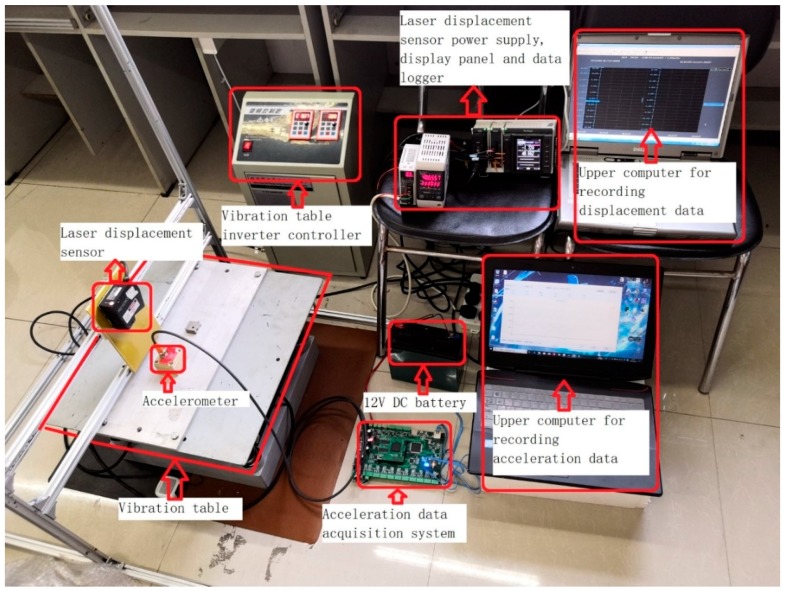
Vibration table physical map.

**Figure 11 sensors-19-03125-f011:**
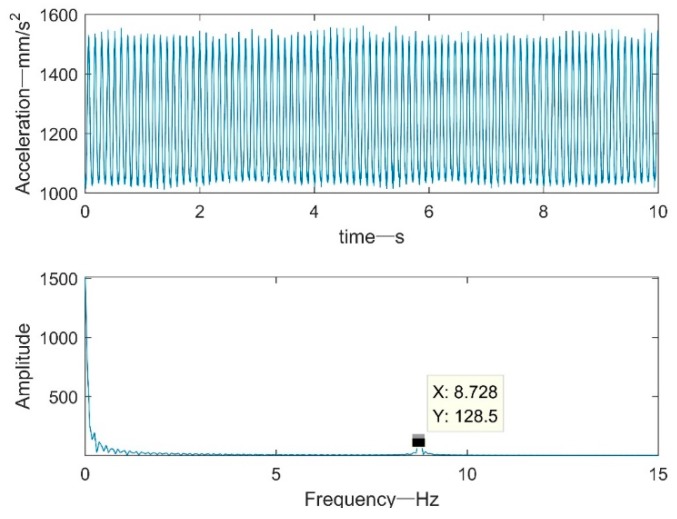
Vibration table vibration acceleration and frequency domain spectrum.

**Figure 12 sensors-19-03125-f012:**
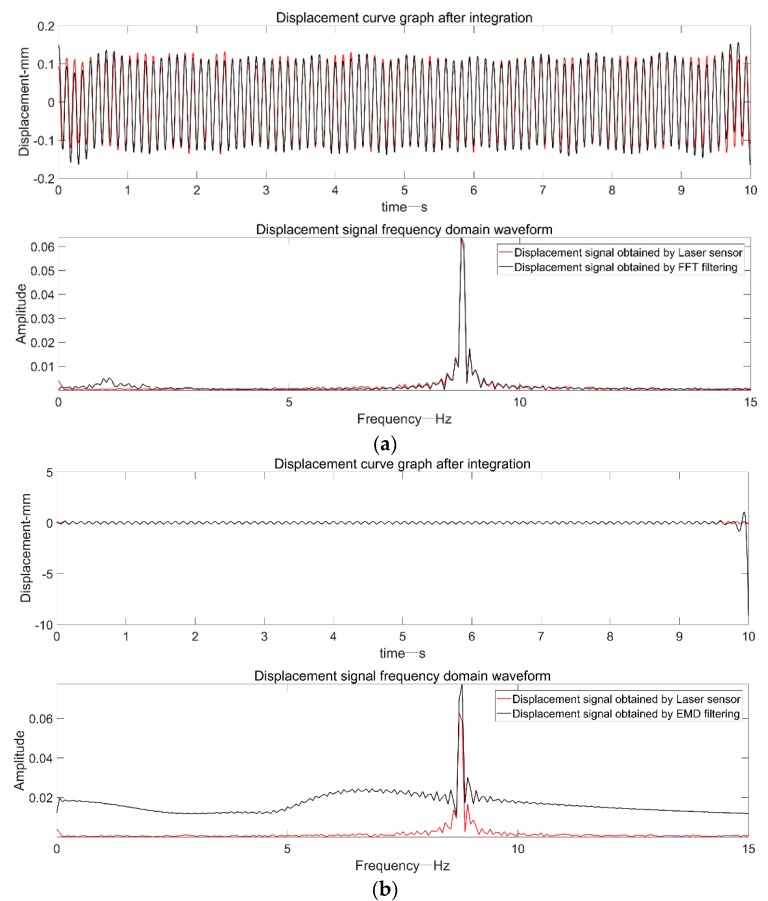
Four methods to realize dynamic displacement reconstruction of a shaking table: (**a**) frequency domain bandpass filtering; (**b**) EMD adaptive filtering; (**c**) FFT+EMD filtering; (**d**) EEMD adaptive filtering.

**Figure 13 sensors-19-03125-f013:**
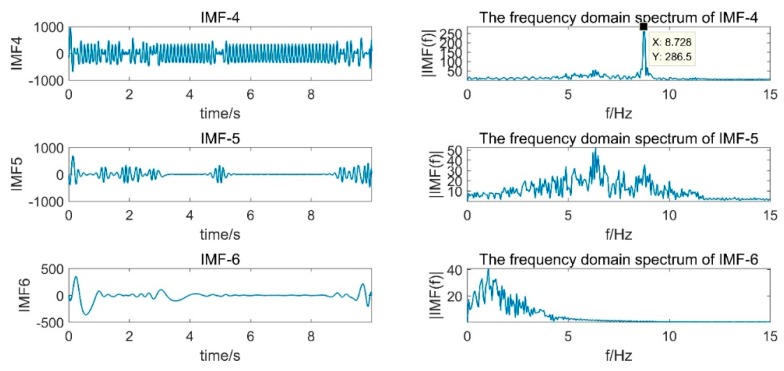
Part of the IMF component obtained by EMD decomposition of the acceleration signal.

**Figure 14 sensors-19-03125-f014:**
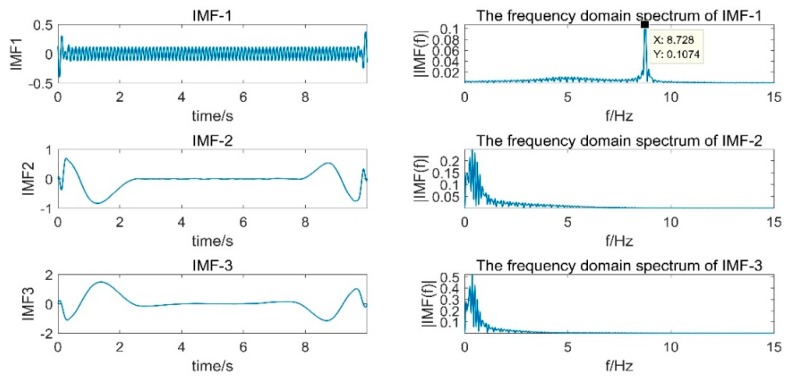
Part of the IMF component obtained by EMD decomposition of the displacement signal.

**Figure 15 sensors-19-03125-f015:**
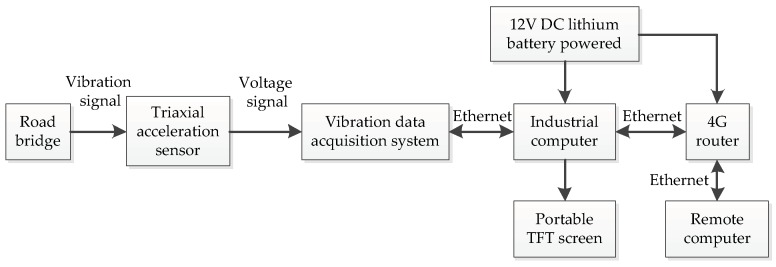
Schematic diagram of bridge site data collection environment.

**Figure 16 sensors-19-03125-f016:**
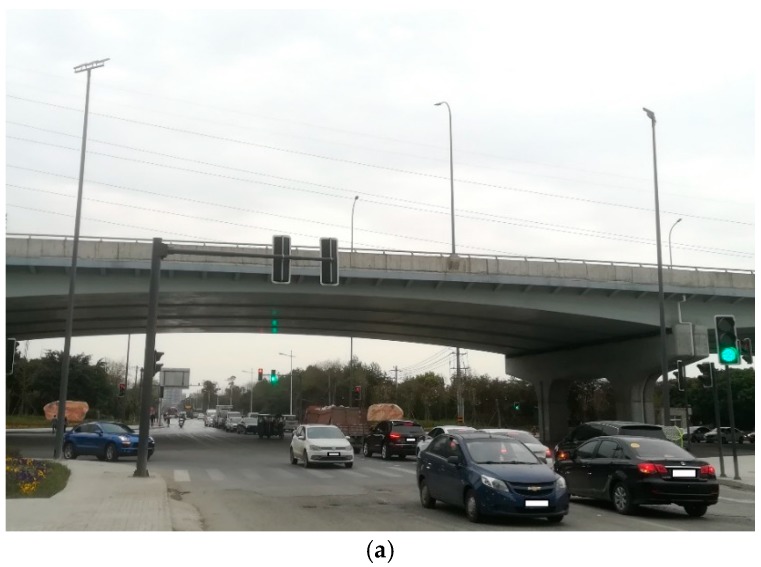
Bridge site data collection physical map: (**a**) Overall view of the bridge; (**b**) Field data collection physical map.

**Figure 17 sensors-19-03125-f017:**
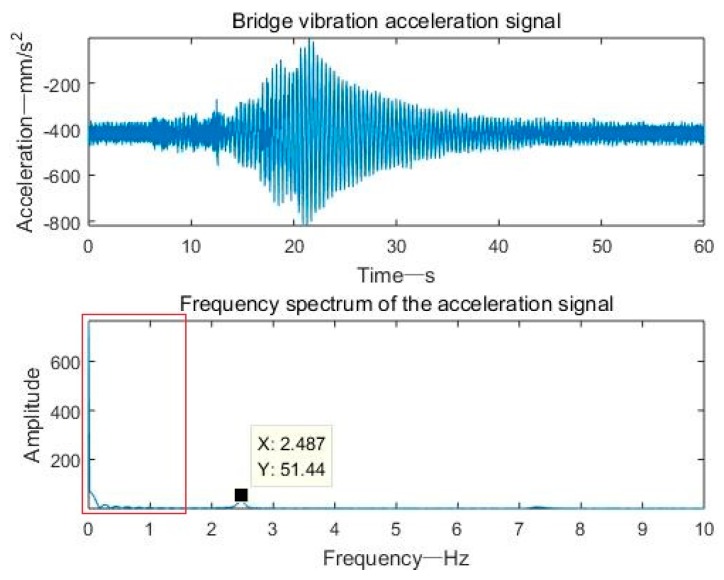
Bridge vibration acceleration signal and frequency domain spectrum.

**Figure 18 sensors-19-03125-f018:**
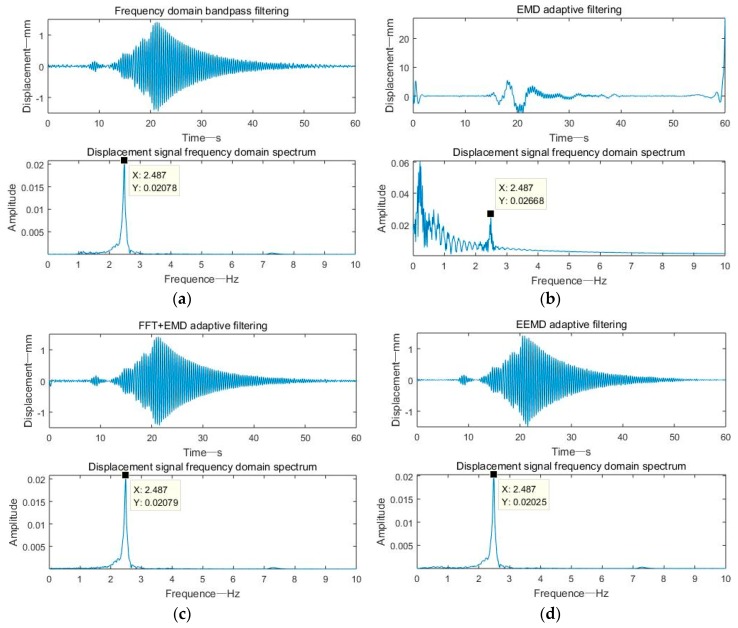
Four methods to realize dynamic displacement reconstruction of bridge: (**a**) Frequency domain bandpass filtering; (**b**) EMD adaptive filtering; (**c**) FFT+EMD filtering; (**d**) EEMD adaptive filtering.

**Figure 19 sensors-19-03125-f019:**
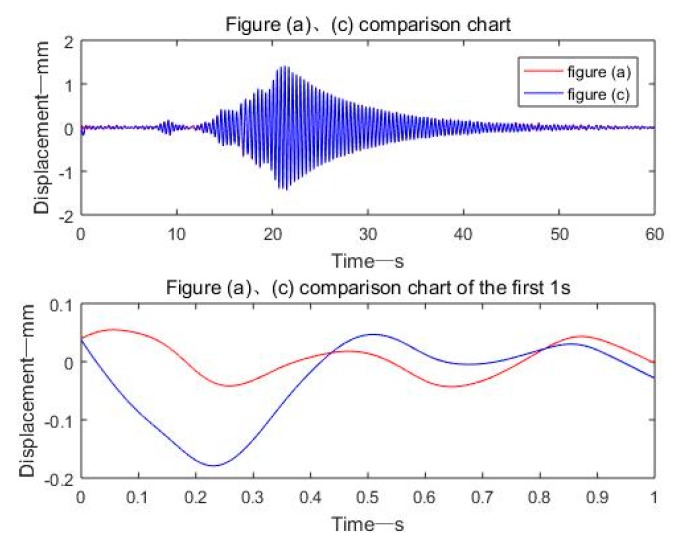
Figure 18a,c displacement signal comparison chart.

**Figure 20 sensors-19-03125-f020:**
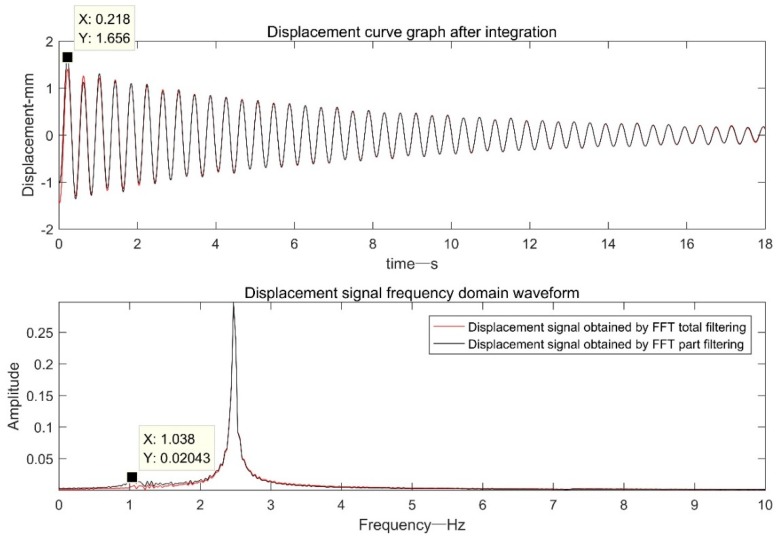
Comparison of “ideal signal” and intercepted signal based on FFT filtering.

**Figure 21 sensors-19-03125-f021:**
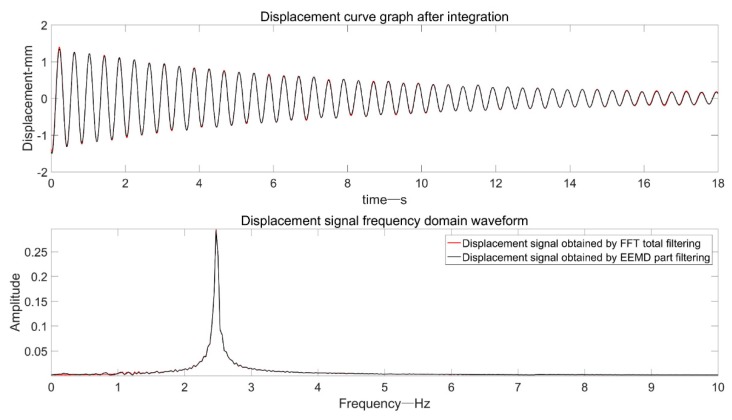
Comparison of “ideal signal” and intercepted signal based on EEMD filtering.

**Table 1 sensors-19-03125-t001:** Displacement reconstruction algorithm.

**Function 1: Calculate the Dynamic Displacement of the Bridge**	
**Input: Acceleration Signal** a(t) **Output: Displacement Signal** x(t)	
**1: for** i=1;i≤N;i++ **do**	15: if $c_{j}$ is an effective IMF then
2: $A_{i}\left(t\right)=a\left(t\right)+n_{i}\left(t\right)$	16: $v_{1}\left(t\right)=v_{1}\left(t\right)+c_{j}$
3: $A_{i}\left(t\right)=\sum_{j=1}^{K}c_{ij}\left(t\right)+r_{iK}\left(t\right)$	17: end if
4: for j=1;j≤K;j++ do	18:$s\left(t\right)=\int_{0}^{t}v_{1}\left(t\right)dt$
5: $c_{j}=\frac{1}{N}\sum_{i=1}^{N}c_{ij}\left(t\right)$	19: for i=1;i≤N;i++ do
6: if $c_{j}$ is an effective IMF then	20: $S_{i}\left(t\right)=s\left(t\right)+n_{i}\left(t\right)$
7: $a_{1}\left(t\right)=a_{1}\left(t\right)+c_{j}$	21: $S_{i}\left(t\right)=\sum_{j=1}^{K}c_{ij}\left(t\right)+r_{iK}\left(t\right)$
8: end if	22: for j=1;j≤K;j++ do
9:$v\left(t\right)=\int_{0}^{t}a_{1}\left(t\right)dt$	23: $c_{j}=\frac{1}{N}\sum_{i=1}^{N}c_{ij}\left(t\right)$
10: for i=1;i≤N;i++ do	24: if $c_{j}$ is an effective IMF then
11: $V_{i}\left(t\right)=v\left(t\right)+n_{i}\left(t\right)$	25: $s_{1}\left(t\right)=s_{1}\left(t\right)+c_{j}$
12: $V_{i}\left(t\right)=\sum_{j=1}^{K}c_{ij}\left(t\right)+r_{iK}\left(t\right)$	26: end if
13: for j=1;j≤K;j++ do	27:$x\left(t\right)=s_{1}\left(t\right)$
14: $c_{j}=\frac{1}{N}\sum_{i=1}^{N}c_{ij}\left(t\right)$	

**Table 2 sensors-19-03125-t002:** Analysis table of integral result error.

	FFT Filtering	EMD Filtering	FFT+EMD Filtering	EEMD Filtering
RMSe/%	5.1636	4.3628	5.4405	1.4678
Ke/%	3.7788	29.3652	7.1083	0.1320
Pe/°	8.7845	23.5410	10.4951	1.9288

**Table 3 sensors-19-03125-t003:** Analysis table of integral result error.

	FFT Filtering	EMD Filtering	FFT+EMD Filtering	EEMD Filtering
RMS/%	3.0499	267.7810	12.5467	0.4658
Ke/%	2.9030	28436	118.8451	0.1337
Pe/°	10.710	70.3518	22.1903	2.1368

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
