# Peer review of "Refactoring and Optimization of Bridge Dynamic Displacement Based on Ensemble Empirical Mode Decomposition"

_sensors, 2019, doi:10.3390/s19143125_

Round 1

Reviewer 1 Report

The paper presents a strategy for displacement estimation based on measurement from MEMS accelerometer using EEMD technique. The paper presents results based on simulation, lab experiments and field experiments and concludes that the EEMD technique gives better results than the FFT based techniques.

The reviewer has the following comments  that need to be addressed before being considered for publication:

1) Firstly the literature review is inadequate. Several other works in the field of acceleration to displacement conversion have been carried out which have not been cited. The work done by Sohn et al. and Chatzi et al, in this area is very interesting. The article makes no mention of Kalman filter based techniques for displacement estimation. So a discussion about those needs to be carried out,

2) The simulation cases and the lab experiment was conducted with a single frequency excitation, it is common practice to have random or white noise excitation to mimic the real conditions in structures and hence is recommended.

3) The parameters used for the EEMD for the noise cancellation have not been stated.how many ensembles were used, level of noise assumed (Why?) etc, Which IMF was used for the integration?

4) The description of the bridge based validation need to be improved as well was the sampling frequency same for the bridge data?

5) The time required for the processing needs to be mentioned and comments made about that, the EEMD is known to be extremely computationally expensive how will large amounts of data be processed for a bridge structure

6) The comparative results for the 4 cases are FFT, EMD, EMD+FFT and EEMD, it seems that the performance is affected by the use of FFT more than the integration step. Also the fft based results may be improved with windowing, overlapping and averaging so comments about this need to be included

7) Also the poor performance of the EMD as compared to the EEMD is obvious due to the averaging step in the EEMD so does not add value. Out of curiosity what are the results for the FFT+EEMD process? 

8) The algorithm in table 1 needs considerable work to ensure clear understanding. There are many symbols, thus providing a list of symbols below the table may be useful.

9) In the discussion about the sampling frequency in equation 7, out of curiosity I will like to ask: will the accuracy improve continuously with increasing sampling frequency or will there be an optimal sampling frequency after which the performance will deteriorate. A similar study for the spatial sampling was carried out by Sazunov et al. for mode shape curvature

10) For bridge structures with closely placed modes, how will the EEMD work? In the presence of other modes being exited how will the decision of IMF be made?

Minor comments

The reference 1- the spelling in the authors is wrong.

It is always a better practice to cite research published in journals than conference proceedings as they are reviewed more thoroughly. An example of relevant publication may be: Soman, R., Kyriakides, M., Onoufriou, T. and Ostachowicz, W., 2018. Numerical evaluation of multi-metric data fusion based structural health monitoring of long span bridge structures. Structure and Infrastructure Engineering14(6), pp.673-684.

Other citations:

Kim, K. and Sohn, H., 2017. Dynamic displacement estimation by fusing LDV and LiDAR measurements via smoothing based Kalman filtering. Mechanical Systems and Signal Processing82, pp.339-355.

Author Response

The reply to the reviewer's comment is in the attachment

Reviewer 2 Report

1.  This paper introduces a method for optimization of bridge dynamic displacement, which has great practical significance.

2.  The writing of the paper needs to be improved to meet the publication standard. A professional editor is recommended for a proof reading and editing. Below are some examples for improving.

3.  Section 2 should be more concise. For example, in Section 2.1, the basic relationship in equations 1-3 can be deleted because they are common knowledge. Equations 4-7 should be rewritten; the subscripts must be clearly marked.

4.       The following lines used to connect two syllables of a word should be deleted.

Line 71 transfor-mation

Line 136 compo-nents

Line 143 com-posed

Line 148 foll-owing (Delete the short line)

5.       Should define cj (Line 152), rK (Line 152), cij (Line 154), riK (Line 154) and H(f) (Line 194). All the symbols in the paper must be defined.

6.       The “if loops” in table 1 are not complete. Explain what need to be done when cj is an effective IMF.

7.       Sizes of figure 12 c) and figure 12 d) should be the same. The quality of most figures should be improved (they are not clear).

8. The statement in Lines 349- 351 is not supported, especially how the fatigue damage was determined? The statement in lines 401-402 is not well supported.

Author Response

The reply to the reviewer's comment is in the attachment.

Round 2

Reviewer 1 Report

I am afraid some of the comments made in the earlier review have not been adequately answered.

For instance the point 2 has not been addressed at all. For the white noise excitation of course one way of validation is through numerical studies. 

Point 3 is not answered completely. how many ensembles were used? 

Point 6: the windowing and overlapping do help in removing the end effects which are seen in all fft based methods and using a band pass filter and claiming it as an rectangular window is not adequate as for FFT studies in most cases Hann window is known to perform the best.

Also the overlap and average effects will definitely improve the noise performance of all the techniques with fft. This is essential as the EEMD's advantage mostly comes from the noise cancellation achieved due to the use of ensembles. so pint 6 needs more careful consideration

Also why was not the displacement measured in the lab setup as a means of obtaining the ideal displacement? without this ideal displacement how can the performance of the 4 techniques be compared?

The response to point 10 shows that the point made by the reviewer was not identified.

For instance in the article below: Larsen, A. (1993). Aerodynamic aspects of the final design of the 1624 m suspension bridge across the Great Belt. Journal of Wind Engineering and Industrial Aerodynamics, 48(2-3), 261–285. doi:10.1016/0167-6105(93)90141-a

There are several different modes present in the band of between 0 and 1 Hz. A situation may occur where there will be more than a few frequencies of vibration which will be in between the 1-10Hz range that is studied here. How will the EEMD based approach work on the reconstruction of such data? Hence in the simulated study the frequency content of the excitation should reflect presence of more than 1 frequency which is more realistic.

So I will encourage the authors to address these comments and compare the performance of the FFT based technique with better implementation.

Author Response

(The authors gave the same response as above.)

Round 3

Reviewer 1 Report

Thank you for the detailed response to my earlier comments.

I am reasonably satisfied with the comparison with frequency based techniques. But I do have questions/concerns  about the applicability of the structure to real applications.

Again the comment 1 was not understood correctly. I was not expecting a white-noise excitation with one peak but something similar to the citation below:

Siringoringo, D. M., & Fujino, Y. (2008). System identification of suspension bridge from ambient vibration response. Engineering Structures30(2), 462-477.

The paper discusses the ambient excitation on a suspension bridge. As can be seen in Figure 8 of the paper above the ambient excitation excites more than one mode of vibration in the freqency range of 1-3Hz. How will be the performance of the EEMD when this kind of excitation occurs on the bridge. The different frequencies of the vibration will be captured by different IMFs and there is a potential for mode mixing.

Comment number 3 corresponds to the details of the parameters chosen for the EEMD algorithms.

needs to be provided in the paper. Number of ensembles is the number of independent noise additions carried out before the average. The level of noise added also affects the performance of the EEMD and needs to be provided in the paper.

About the application on the vibration table, the experimental limitations are understood but some other setup can be used for the estimation. Use of displacement measurement can be achieved by LVDT or other sensors which perform well in the laboratory and are not overly expensive. Also the excitation for broadband of frequencies can be easily realized through hammer excitation. So this needs to be performed as well.

Was vibration analysis carried out on the bridge? The reference to these studies needs to be provided. Also the description of the bridge needs to be expanded. If possible with name, type of construction etc. The presence of the single frequency peak in the 1-10 Hz range is surprising to the reviewer. In his experience multiple closely spaced frequency modes are observed in the bridge. So having a description of the bridge will help.

Also the comparison provided in Figure 20 is trivial. In the absence of excitation the noise in the measurement is dominant. So by performing the integration step the measurement noise effects get amplified. The EEMD due to its excellent performance for overcoming noise will obviously perform better.

Also can the authors expand on why the other methods perform poorly towards the end of the signal? are there any systematic errors in the other methods?

So in nutshell, I am reasonably convinced with the methodology and its performance for the numerical simulations but more work needs to be provided for the laboratory and field validations. Hence I will like to see some major changes in the sections 3.2 and onwards.

Author Response

(The authors gave the same response as above.)

Round 4

Reviewer 1 Report

The reviewer is impressed by the speed at which additional experiments were performed as well as the detail in which my comments were answered. Most of the responses are satisfactory.

The details of the white noise added in the simulated example have not been provided. What was the level of noise (SNR) etc needs to be provided. Also the details of the EEMD given in the response to reviewer's comments should be included in the manuscript. In the eyes of the reviewer 40% noise is a large unrealistic assumption. This may lead to additional wrong peaks in the analysis and should be used with care.

Also I will like the authors to edit the conclusions by mentioning that the techniques are not suitable for the broadband excitation as shown by the researchers in the comments. 

The conclusion can also be expanded with plans for future work and other limitations and scope of the methodology.
